# Measles Virus-Based Vaccine Expressing Membrane-Anchored Spike of SARS-CoV-2 Inducing Efficacious Systemic and Mucosal Humoral Immunity in Hamsters

**DOI:** 10.3390/v16040559

**Published:** 2024-04-03

**Authors:** Zhi-Hui Yang, Yan-Li Song, Jie Pei, Song-Zhuang Li, Rui-Lun Liu, Yu Xiong, Jie Wu, Yuan-Lang Liu, Hui-Fen Fan, Jia-Hui Wu, Ze-Jun Wang, Jing Guo, Sheng-Li Meng, Xiao-Qi Chen, Jia Lu, Shuo Shen

**Affiliations:** Wuhan Institute of Biological Products Co. Ltd., Wuhan 430207, China; 13552116951@163.com (Z.-H.Y.); 13001216531@126.com (Y.-L.S.); pj930608@163.com (J.P.); li_songzhuang@163.com (S.-Z.L.); lrl15549793633@163.com (R.-L.L.); 13476165881@163.com (Y.X.); wujie20230805@126.com (J.W.); songyanlisyl@gmail.com (Y.-L.L.); qingdaoaoyu@163.com (H.-F.F.); jhui_w2000@163.com (J.-H.W.); wangzejun@sinopharm.com (Z.-J.W.); guojingwh2004@hotmail.com (J.G.); mengshengli@sinopharm.com (S.-L.M.); chenxiaoqi@sinopharm.com (X.-Q.C.)

**Keywords:** SARS-CoV-2, Omicron BA.2, spike protein, measles virus, SP-D, trimerization tag, mucosal immunity

## Abstract

As SARS-CoV-2 continues to evolve and COVID-19 cases rapidly increase among children and adults, there is an urgent need for a safe and effective vaccine that can elicit systemic and mucosal humoral immunity to limit the emergence of new variants. Using the Chinese Hu191 measles virus (MeV-hu191) vaccine strain as a backbone, we developed MeV chimeras stably expressing the prefusion forms of either membrane-anchored, full-length spike (rMeV-preFS), or its soluble secreted spike trimers with the help of the SP-D trimerization tag (rMeV-S+SPD) of SARS-CoV-2 Omicron BA.2. The two vaccine candidates were administrated in golden Syrian hamsters through the intranasal or subcutaneous routes to determine the optimal immunization route for challenge. The intranasal delivery of rMeV-S+SPD induced a more robust mucosal IgA antibody response than the subcutaneous route. The mucosal IgA antibody induced by rMeV-preFS through the intranasal routine was slightly higher than the subcutaneous route, but there was no significant difference. The rMeV-preFS vaccine stimulated higher mucosal IgA than the rMeV-S+SPD vaccine through intranasal or subcutaneous administration. In hamsters, intranasal administration of the rMeV-preFS vaccine elicited high levels of NAbs, protecting against the SARS-CoV-2 Omicron BA.2 variant challenge by reducing virus loads and diminishing pathological changes in vaccinated animals. Encouragingly, sera collected from the rMeV-preFS group consistently showed robust and significantly high neutralizing titers against the latest variant XBB.1.16. These data suggest that rMeV-preFS is a highly promising COVID-19 candidate vaccine that has great potential to be developed into bivalent vaccines (MeV/SARS-CoV-2).

## 1. Introduction

The outbreak and prevalence of severe acute respiratory syndrome coronavirus 2 (SARS-CoV-2) has caused immense loss of human life and global socioeconomic stability [1]. Up to now, there have been approximately 771 million reported cases worldwide, resulting in nearly 6.97 million deaths (with a mortality rate of approximately 0.90%). The primary symptoms of SARS-CoV-2 infection are respiratory, although there has been a rising incidence of other symptoms, such as cognitive impairment. Several SARS-CoV-2 vaccines based on adenovirus vector, mRNA, recombinant S protein, and inactivated virus methods have been approved for vaccination in humans over 5 years old [2]. Although the current SARS-CoV-2 vaccine has achieved great success, some limitations remain. The protection provided by the current vaccines began to decline after 3 months [3], which requires a third or fourth dose of vaccine to enhance the level of neutralizing antibodies. Moreover, the current vaccines have extremely limited cross-protection efficacy against recently emergent SARS-CoV-2 variants of concern (VoCs) [4,5]. Large amounts of evidence have shown that vaccine-induced NAbs were significantly weakened or even failed to neutralize VoCs such as Omicron BA.2, which has been reported to severely dampen antibody neutralization [6,7]. In addition, early studies reported the markedly reduced neutralizing activity against Omicron BA.2 of convalescent or BBIBP-CorV vaccination sera compared with the WT virus [8].

There is an urgent need for a safe and effective SARS-CoV-2 vaccine to halt the current epidemic. Spike protein has been proven to be immunogenic for many CoVs, and the S gene is regarded as a key target for SARS-CoV-2 vaccines. The native S proteins in the virion are in their “prefusion” form and exist as a trimeric form on the surface of the virus and are shown to have higher immunogenicity but not the monomeric form [9,10,11,12]. The prefusion forms of glycoprotein have been reported to be more potent in inducing NAbs than their post-fusion forms for HIV, respiratory syncytial virus (RSV), and SARS-CoV-2 [13,14,15]. In its native state, spike protein is a trimer, and the recombinant trimeric full-length spike proteins have been reported to be highly immunogenic and able to elicit an efficacious protective immune response [9,16]. The full-length spike proteins contain a transmembrane domain (TM) involved in mediating trimerization [17]. However, when TM is deleted, spike proteins will exist as a monomer with poor immunogenicity, making it impractical in the production of trimeric, recombinant vaccines.

To solve the problem, we have developed a neotype platform technology named SP-D trimer-tag, similar to Trimer-Tag, for rapidly producing a prefusion form of trimeric spike proteins. Human pulmonary surfactant protein D (SP-D) is a member of the collectin family [18]. The secretion form of SP-D is composed of four trimer subunits, including the N-terminal domain, collagen domain, neck domain, and C-terminal carbohydrate recognition domain (CRD). Early studies show that the neck plus CRD domains can drive the trimerization of homologous and heterologous proteins [19,20].

Live attenuated measles virus vaccine (MeV) has been widely administrated to infants and children since the 1960s and is universally acknowledged as one of the safest and most effective vaccines [21]. MeV is an enveloped, single-stranded, negative-strand RNA virus belonging to the genus *Morbillivirus* in the family *Paramyxoviridae*. Since the establishment of the reverse genetics system of MeV in 1995, MeV has been used as a vector to deliver vaccines against more than 50 human pathogens [22,23]. Among them, the Zika virus, Lassa virus, and HIV vaccines based on MeV vectors are undergoing phase I clinical trials. In addition, recent human clinical trials have demonstrated that it is safe and highly immunogenic in healthy adults with pre-existing antibodies to MeV.

In this study, we developed two MeV-based vaccine candidates expressing chimeric stabilized prefusion spike proteins (preFS) and secretory trimeric spike proteins (S+SPD) derived from the Omicron BA.2 viral sequence and evaluated their immunogenicity and protective effect against Omicron BA.2 challenge in a golden Syrian hamster model.

## 2. Materials and Methods

### 2.1. Ethical Approval

The golden Syrian hamsters in the study were obtained from the Animal Ethics Committee (AEC) of the Wuhan Institute of Biological Products and were housed within individual ventilated cages (IVC). All procedures of the experiments were executed according to the guidelines and regulations issued by the Standardization Administration of China. For anesthesia and euthanasia, the golden Syrian hamsters were subjected to 3% isoflurane and CO_2_ inhalation, respectively.

### 2.2. Plasmid

A stabilized prefusion Omicron BA.2 full-length S (preFS) was synthesized with the removal of the furin cleavage site and two proline mutations (2P mutation, K986P, and V986P). Furthermore, the modified S without transmembrane domain and cytoplasmic tail and with a human pulmonary surfactant protein D (S+SPD) was synthesized. Both fragments were amplified by PCR and inserted into pMeV-hu191 between H and L genes using two restriction enzymes, *Bsi*WI and *Bss*HII. The two plasmids were designated as pMV3-preFS and pMV3-S+SPD, respectively. The two constructs were confirmed by sequencing. The genes of the preFS gene and S+SPD used in this study were condon-optimized for the green monkey kidney cells expression.

### 2.3. Recovery of MeV-hu191 Recombinants Expressing Omicron BA.2 S Antigens

BSR-T7/5 cells stably expressing T7 polymerase were seeded onto 6-well plates at 1 × 10^6^ cells per well in duplicates [24]. pMV3-preFS, pMV3-S+SPD, and support plasmids encoding the MeV-hu191 ribonucleocapsid complex (pcDNA-N/P/L) were co-transfected into BSR-T7/5 cells on the next day. On day 5 post-transfection, cells and supernatants were collected and transferred to T25 flasks seeded with Vero cells. When syncytia reached 90% of the confluent monolayers, the recovered measles virus recombinants were harvested and passaged in Vero cells. The viral titer was determined by a CCID_50_ assay in Vero cells.

### 2.4. Multi-Step Growth Curves

The virus growth kinetics of rMeVs were studied on confluent monolayers of Vero cells in six-well plates. Vero cells were infected with individual viruses at a multiplicity of infection (MOI) of 0.001. After 2 h of absorption, the inoculum was removed. The cells were washed three times with DMEM, fresh DMEM was added, and the plates were placed at temperatures of 34 °C and 37 °C, respectively. The cell culture media were obtained at intervals of 24 h, and virus titers were examined and expressed as TCID_50_/mL.

### 2.5. Plaque Assay

The proliferative capacity of rMeV, rMeV-preFS, and rMeV-S+SPD was determined by a plaque assay on Vero cells in 6-well plates, respectively. Confluent Vero cells in 6-well plates were infected with a serial 10-fold dilution of the recombinants in DMEM. After absorption for 1 h at 37 °C, cells were washed three times with DMEM and overlaid with 2 mL of DMEM containing low-melting agarose (1% *w*/*v*). Six days post-infection, cells were fixed with paraformaldehyde (4%, *w*/*v*) for 2 h and stained with crystal violet (1% *w*/*v*). The diameter of the plaques was measured with the software Nano Measurer 1.2.

### 2.6. Detection of Omicron BA.2 Spike Protein Expression in Infected Cells by Indirect Immunofluorescence Assays

Monolayers of Vero cells in a 6-well plate were infected with parental rMeV, rMeV-preFS, and rMeV-S+SPD-expressing S antigens as described above at an MOI of 0.001, cultured for 3 days at 34 °C. When syncytia were visible, the supernatants were replaced with cold acetone (80% *v*/*v*) for 30 min at −20 °C. Then, cells were incubated for 1 h at 37 °C with anti-SARS-CoV-2 spike chimeric mAb (Sino Biological, Beijing, China) diluted to 1 in 1000. The cells were stained for 45 min at 37 °C with secondary antibodies against human IgG (H+L) and Alexa Fluor 488 (Invitrogen, Waltham, MA, USA) at a dilution of 1 in 500. After subsequent incubation with a DAPI staining solution diluted to 1 in 1000 (Beyotime, Shanghai, China), fluorescent signals in cells were analyzed under an Axioplan 2 epifluorescence microscope (Zeiss, Oberkochen, Baden-Württemberg, Germany). Photographs were taken with an Axiocam MRm camera and processed with the Axiovision software (Version 4.2, Zeiss).

### 2.7. Analysis of Omicron BA.2 S Antigen Expression by Western Blotting

Vero cells were infected with rMeVs as described above. At the indicated times post-infection, the supernatants were harvested and kept at −80 °C. Infected cells were lysed in SDS lysis buffer (Beyotime, Shanghai, China) containing protease inhibitor cocktail (Beyotime, Shanghai, China) and frozen at −80 °C. Samples were mixed with 4 × Laemmli sample buffer (Bio-Rad, Hercules, CA, USA) added with BME and 4 × Protein Native PAGE Loading Buffer without BME (TaKaRa, Kyoto, Japan), respectively. The mixtures were treated at 100 °C for 10 min. Samples were subjected to SDS-PAGE using SurePAGE-pre-cast 8% gel with MOPs running buffer (Genscript, Nanjing, China), and proteins were transferred to nitrocellulose membrane (Amersham, London, UK) in an eBlot L1 system (Bio-Rad, Hercules, CA, USA). The blot was split into three parts, then probed with mouse anti-Omicron RBD antibody at a dilution of 1/5000 (Sino Biological, Beijing, China), rabbit anti-MeV-N3 antisera at a dilution of 1 in 5000, and β-tubulin mAb at a dilution of 1 in 5000 (Epizyme, Shanghai, China), respectively. Following subsequent incubation with horseradish peroxidase-conjugated goat anti-mouse/rabbit IgG secondary antibody (Boster, Wuhan, China) at a dilution of 1 in 100,000, the blot was developed with an Immobilon Western HRP substrate (Millipore, Burlington, MA, USA) and exposed to Amersham ImageQuant 800 (Amersham, London, UK).

### 2.8. Sucrose Gradient Fractionation

Supernatants of 3 × 10^7^ infected cells were clarified by centrifugation at 8000× *g* for 30 min and then layered carefully on top of a 10, 20, 30, 40, 50, and 60% (5 mL each) sucrose-0.01 M PBS step gradient, followed by ultracentrifugation in an SW32 rotor at 15,093 × *g* for 4 h at 4 °C. To determine the density and content, the bottom of the tube was punctured and 1.5 mL of each fraction was collected. A total of 24 fractions were stored at −80 °C. Each fraction was mixed with 4 × Laemmli sample buffer with BME, and 4 × Protein Native PAGE Loading Buffer was added. The electrophoresis and protein staining were performed as described above.

### 2.9. Immunization and Challenge Studies in Golden Syrian Hamsters

Groups of four-week-old female golden Syrian hamsters (n = 8) were subcutaneously or intranasally mock-inoculated with DMEM medium or inoculated with rMeV-preFS, rMeV-S+SPD, and parental rMeV at a dose of 1 × 10^6^ TCID_50_ in 100 μL per hamster twice in three weeks. For the intranasal vaccination study, hamsters were anesthetized with isoflurane before inoculation of viruses. At weeks 2, 4, and 5 post-priming, blood was collected from each hamster for antibody detection. At week 3, for post-booster immunization, five animals in each group were randomly chosen, transferred into the BSL-3 facility, and challenged with Omicron BA.2 at a dose of 1 × 10^6^ TCID_50_ per animal. After the challenge, the body weight of each hamster was monitored daily. At day 5 post-challenge, all hamsters were euthanized. Left lung and nasal turbinate were collected for detection of infectious Omicron BA.2 and viral RNA. In addition, the right lung was fixed with 4% (*w*/*v*) paraformaldehyde for histology and immunohistochemistry (IHC).

### 2.10. Detection of Omicron BA.2 Spike-Specific IgG and IgA Antibodies by ELISA

Briefly, 96-well plates were coated with BA.2 spike S1+S2 trimer protein (Sino Biological, 40589-V08H28) at 0.5 μg/mL in 0.01 M PBS (100 μL/well) overnight at 4 °C. The plates were washed three times with PBST (0.01 M PBS containing 0.1%Tween-20) and blocked with blocking buffer (3% BSA in PBST, 200 μL/well) for 2 h at 37 °C. Hamster serum samples were 2-fold serially diluted in blocking buffer and added to spike protein-coated plates (100 μL/well) followed by a 2 h incubation period at 37 °C. Then, the plates were washed four times with PBST and incubated with HRP-conjugated goat anti-Syrian hamster IgG (Abcam, ab6892) at a dilution of 1: 10,000 (100 μL/well) for 1 h at 37 °C. After washing, antibody binding was detected by the addition of TMB substrate (100 μL/well) at 37 °C for 15 min, and the reaction was stopped with 2 M H_2_SO_4_ (50 μL/well). The optical densities (ODs) were recorded at 450 and 630 nm using a BioTek microplate reader. Endpoint titers were determined as the reciprocal of the highest dilution that had an absorbance value of 2.1-fold greater compared with that of the background level of the normal control serum. Antibodies titers were reported as geometric mean titers (GMTs).

For IgA in bronchoalveolar lavage fluid (BALF), nasal lavage (NAL) and fecal samples were tested as above with minor modifications. In brief, the plates were blocked with 3% BSA in PBST for 2 h at 37 °C. The samples were incubated at 37 °C for 2 h. IgA antibodies were detected with HRP-conjugated rabbit anti-hamster IgA diluted to 1 in 1000 (Brookwood Biomedical, Jemison, AL, USA, sab3003A).

### 2.11. Detection of SARS-CoV-2 (Omicron BA.2, Prototype Strain, and XBB.1.16) and Measles Virus Neutralizing Antibodies by Microneutralization Assays

SARS-CoV-2 strains or rMV3-eGFP (constructed in early research [24]) were diluted in DMEM so that 50 μL of virus suspension contained 100 CCID_50_. Equal volumes of 2-fold, serial dilutions of individual sera were added to virus suspensions (2 duplicates for each dilution) in 96-well plates. After incubation for 2 h at 37 °C, 6 × 10^4^ Vero cells in 100 μL of DMEM containing 10% FBS were added. Cells and negative serum controls were included, and virus back-titration was performed to make sure that the virus titers were in the range of 32 to 320 CCID_50_/50 μL. After incubation for three days for measle viruses and seven days for SARS-CoV-2 strains at 37 °C, the neutralizing titers were calculated using the Reed–Muench method by observing CPE or fluorescent signals of eGFP using CTL ImmunoSpot^®^S6 and were defined as the reciprocal of the highest dilution at which CPE or fluorescent signals in 50% of the wells was completely inhibited compared with negative serum-treated viruses.

### 2.12. Measurement of Omicron BA.2 Genomic RNA Burden by Quantitative Real-Time PCR (qRT-PCR)

The viral RNA from homogenized left lung and nasal turbinate were extracted using an RNA extracting kit (Daangene, Guangzhou, China). The qRT-PCR was performed with an Applied Biosystems 7500 Real-Time PCR System (Invitrogen, Waltham, MA, USA). A standard curve was generated using a plasmid encoding the SARS-CoV-2 nucleocapsid (N) gene or ORF 1ab gene conserved region to calculate the copy numbers of viral RNA in the left lung or nasal turbinate of a challenged hamster as described previously [15].

### 2.13. Histopathology and Immunohistochemistry (IHC)

A total of 4% paraformaldehyde-fixed, paraffin-embedded right lungs obtained from hamsters were cut into 5 μm-thin sections and stained with hematoxylin and eosin (H&E) for examination of histological changes. All sections were evaluated and scored by a staff from the Emerging Infectious Diseases Laboratory that was blinded to the vaccination groups. For immunohistochemistry, the sections were counterstained using hematoxylin and stained with an Anti-SARS-CoV-2 N protein at a dilution of 1:50 (Abcam, Cambridge, MA, USA). The slides were scanned using an OptraSCAN scanner.

### 2.14. Statistical Analysis

The statistical analysis was performed using GraphPad Prism 8.3.0. Statistical significance was indicated as follows: ns, not significant (*p* ≥ 0.05); *, 0.01 ≤ *p* < 0.05; **, *p* < 0.01; ***, *p* < 0.001; and ****, *p* < 0.0001. For multiple comparisons, the statistical difference was determined using one-way ANOVA or two-way ANOVA tests. To compare the two groups, *t*-tests were used.

## 3. Results

### 3.1. Rescue and Characterization of rMeVs Expressing Prefusion Spike Proteins of SARS-CoV-2 Omicron BA.2

To better understand the experimental setup and phases, an experimental flow chart is shown in Figure 1A. The full-length prefusion-stabilized S in rMeV-preFS can be viral membrane-anchored, while the ectodomain of S in rMeV-S+SPD were linked to the human pulmonary surfactant protein-D trimerization motif, replacing the transmembrane domain and cytoplasmic tail (S+SPD) (Figure 1B,C). The two S-recombinants and the parental strain were rescued and purified by plaque assays. The viral genomic RNAs were amplified by RT-PCR. Sequencing of the PCR products confirmed that the recombinants with preFS and S+SPD insertions were correctly constructed and rescued.

To evaluate the growth characteristics of the rescued recombinants, the plaque-forming ability of the rescued viruses was first recorded. Both rMeV-preFS and rMeV-S+SPD formed smaller plaques on Vero cells than that of the parental rMeV (Figure 1D). Indeed, the average plaque diameters of the two recombinants with S insertion were significantly smaller (1.08 ± 0.15 and 1.08 ± 0.18 mm) than that of the parental rMeV (1.40 ± 0.19 mm, Figure 1E). Secondly, propagation of the recombinants was compared at different temperatures. Multistep replication curves at both 34 °C and 37 °C showed that these two recombinants with S insertion had similar replication kinetics to that of the parental virus (Figure 1F,G). However, the two S recombinants, like the parental virus, achieved higher titers at 34 °C than that at 37 °C (Figure 1H). The titers in supernatants and cell pellets showed that the ratio of extracellular to intracellular viruses was 1:3 at 34 °C for three viruses (Figure 1I). The syncytia formation by recombinant viruses at 34 °C was significantly delayed, consistent with the results of the plaque size assays (Figure 1J). These data suggested that the insertion of preFS and S+SPD genes into the junction between the H and L genes in the MeV genome might attenuate the recombinants by slowing their replication but did not significantly disturb the propagation and affect their final viral titers in cell culture. Obviously, the optimal culture temperature of the virus was determined to be 34 °C.

### 3.2. Omicron BA.2 S Antigens Were Highly Expressed by rMeV Vector

To examine the expression of Omicron BA.2 spike proteins in cells infected with the recombinants, IFA was conducted using an anti-SARS-CoV-2 RBD chimeric mAb. The results revealed a robust expression of both preFS and S+SPD protein, whereas no signals were observed in cells infected with parental rMeV (Figure 2A). The cell culture supernatants and cell lysates were harvested at 96 h post-infection and analyzed by Western blot using anti-Omicron RBD monoclonal antibody (MM117), rabbit MeV-N3 antisera, and β-tubulin antibody, respectively. Without treatment with BME, S monomer (with molecular weights of about 200 kDa), S dimer, and S trimer were detected in the rMeV-preFS-infected cells (Figure 2B). However, only two protein bands at the same positions of S monomer and S trimer were detected in rMeV-S+SPD-infected cells (Figure 2B). As expected, in rMeV-preFS-infected or rMeV-S+SPD-infected cells, the 250 kDa uncleaved, stabilized preFS or S+SPD proteins (S monomer) were detected, and no cleaved S proteins were detected as the cleavage site of S1/S2 was removed.

Furthermore, expression time courses were performed in Vero cells infected with the two S recombinants. For rMeV-preFS, the S protein expression had greatly increased on day 3 and reached the peak on day 5 post-infection, detected in both cell lysates and supernatants together with the N protein of the measles virus (Figure 2C). The results indicated that the S proteins assembled into viral particles. For rMeV-S+SPD, the S proteins were mainly detected in the supernatants, but only a few in cell lysates, and reached the highest expression level on day 5 post-infection (Figure 2D). The expression of spike proteins from rMeV-S+SPD-infected cells in supernatants were higher than that in cell lysates at all time points (Figure 2D). The N protein of measles virus from rMeV-S+SPD-infected cells were mainly detected in cells lysates, which is the same as the rMeV-preFS-infected cells (Figure 2D). These results demonstrated that Omicron BA.2 S proteins were highly expressed by rMeV-preFS and rMeV-S+SPD. As expected, preFS assembled into virion and S+SPD secreted out of cells and both S proteins formed trimers. It was noticed that double bands of monomers of preFS and S+SPD were only detected in cell lysates but not in supernatants, indicating that only mature and fully glycosylated S proteins appeared in virions or were secreted in supernatants.

### 3.3. The Spike Proteins Expressed by rMeV-preFS and rMeV-S+SPD Were Successfully Incorporated into Measles Virions and Secreted

To further demonstrate that the two forms of spike proteins were associated with MeV virions and secreted, the recombinant viruses in supernatants were subsequently fractionated by ultracentrifugation through a 10–60% sucrose gradient. It was expected that SARS-CoV-2-Spike proteins with TM and CT would be integrated into the lipid envelope of the matured recombinant virus. On the contrary, the spike proteins derived from rMeV-S+SPD-infected cells would be detected mainly in the supernatant of the culture medium in the form of secretory proteins. Therefore, cell-free virus particles from supernatants of rMeV-preFS- or rMeV-S+SPD-infected Vero cells were sedimented to equilibrium on a 10–60% sucrose gradient by ultracentrifugation. To determine the viral proteins in each fraction, gradient fractions were analyzed by Western blotting and probed with MM117 for SARS-CoV-2 spike proteins, rabbit anti-MeV-N3 antisera for N proteins, and anti-MeV-P monoclonal antibody for P proteins of measles viruses (Figure 3). The N and P proteins in virus particles were detected in fractions 18–22 of rMeV-preFS (Figure 3A) and rMeV-S+SPD in the density range of 1.15 to 1.20 g/cm^3^ (Figure 3B). The buoyant densities of all particles did not show differences between recombinant MeVs and wild-type MeV (in the range of 1.16 to 1.20 g/cm^3^) [25]. However, full-length preFS spike proteins were co-located with measles virus particles (Figure 3A), but S+SPD proteins were not and appeared in fractions 3 to 8 (Figure 3B). It was evident that the full-length preFS spike proteins in the supernatant of the rMeV-preFS-infected cell were dominantly detected in fractions 16–22 (1.13–1.20 g/cm^3^). Furthermore, the spike proteins from the rMeV-S+SPD-infected cells were only detected in fractions 1–10 (1.00–1.07 g/cm^3^), but not in the fractions of virus particles. These data clearly demonstrated that the spike proteins expressed from the rMeV-preFS-infected cells were indeed incorporated into measles virions, while those from rMeV-S+SPD-infected cells were secreted out of cells.

### 3.4. Immunization with rMeV-preFS Induced a Strong Mucosal IgA Response

Recent studies have shown that mucosal secretory IgA antibody plays an important role in protecting infection against SARS-CoV-2 and its variants [26,27,28,29]. However, the optimal route for MeV-based vaccine immunization to induce mucosal immune response is uncertain. Thus, we next compared the mucosal humoral immunogenicity through subcutaneous and intranasal routes for the two MeV-based vaccines. Six four-week-old golden Syrian hamsters in each group were first primed with 1 × 10^6^ CCID_50_ of the rMeV-preFS, rMeV-S+SPD, or parental rMeV via subcutaneous and intranasal routes, respectively, and boosted with the same doses 21 days later (Figure 4A). The S-specific serum IgA titers in the intranasal group immunized with rMeV-preFS or rMeV-S+SPD reached similar levels on days 14, 28, and 35 compared with the subcutaneous group (Figure 4B). Interestingly, S-specific serum IgA titers of the rMeV-preFS-immunized group were higher than those of the rMeV-S+SPD-immunized group at the same time points regardless of intranasal or subcutaneous immunization (Figure 4B).

Importantly, it was found that all hamsters in the rMeV-preFS-immunized group had similar levels of S-specific IgA antibody in the BALF and NAL on days 28 and 35 through subcutaneous and intranasal immunization (Figure 4C,D). Only two of the six hamsters in the intranasally rMeV-S+SPD-immunized group produced an extremely low level of S-specific IgA antibody on day 14, but relatively high levels of IgA antibody were detectable on days 28 and 35 in all hamsters in the BALF and NAL (Figure 4C,D). Only the extremely low level of S-specific IgA antibody in the BALF (Figure 4C) and NAL (Figure 4D) in the partial subcutaneously rMeV-S+SPD-immunized group were detected at these time points. The S-specific IgA antibody in the BALF of the intranasally rMeV-S+SPD-immunized group was significantly higher than those of the subcutaneously immunized group on days 28 (*p* < 0.001) and 35 (*p* < 0.0001) (Figure 4C). Moreover, the S-specific IgA antibody in the NAL of the intranasally rMeV-S+SPD-immunized group was also significantly higher than those of the subcutaneous group on days 28 (*p* < 0.01) and 35 (*p* < 0.0001) (Figure 4D). It was observed that the minimal S-specific IgA in feces was only detected in the rMeV-preFS-immunized group and had no significant difference between the intranasal group and subcutaneous group at all time points (Figure 4E).

These data indicate that intranasal delivery of MeV-based vaccines can induce more robust mucosal immune response and that the effect of rMeV-preFS is better than rMeV-S+SPD.

### 3.5. rMeV-preFS Was Highly Immunogenic in Golden Syrian Hamsters via the Intranasal Route

To evaluate the humoral immunogenicity of recombinants expressing the preFS or S+SPD as vaccine candidates against SARS-CoV-2 variant Omicron BA.2, a proof of principle study was performed in golden Syrian hamsters that are suitable for evaluating the efficacy of COVID-19 vaccine candidates or antiviral drugs [30,31]. Groups of eight four-week-old golden Syrian hamsters were primed at a dose of 1 × 10^6^ CCID_50_ of the rMeV-preFS, rMeV-S+SPD, or parental rMeV and boosted on day 21 post-priming (Figure 5A). The animals had no fever (*p* ≥ 0.05), and the body weights of hamsters remained constant compared with those of the mock-immunized control group (Figure 5B,C), indicating well-tolerated post-immunizations. High IgG antibody titers were detected in the sera of all eight hamsters in the rMeV-preFS group on day 14 after a single-dose vaccination (Figure 5D). After a booster immunization on day 21, antibody titers dramatically increased on day 28 and remained at similar levels on days 35 and 42 (Figure 5D).

The NAbs triggered by vaccination are believed to be important for vaccine-induced protection against SARS-CoV-2 infection [32]. Therefore, the level of neutralizing antibodies against Omicron BA.2, the prototype strain, or the latest epidemic, the XBB.1.16 strain at the time of experiments, induced by rMeV-preFS or rMeV-S+SPD in hamsters were detected by a microneutralization assay. The NAbs titer against Omicron BA.2 of sera from the rMeV-preFS group was significantly higher than the titers of sera collected from the rMeV-S+SPD group (*p* < 0.0001) (Figure 5E). High NAbs titers were detected in the rMeV-preFS group (GMT = 136.6) on day 14 after a single-dose vaccination and greatly increased on days 28 (GMT = 2982.9) and 35 (GMT = 4069) after a booster immunization on day 21 (Figure 5E). However, the NAbs titers of serum from the rMeV-S+SPD-immunized hamsters were only detectable on days 28 (GMT = 57.5) and 35 (GMT = 75.6) after a booster immunization (Figure 5E). The GMTs against the prototype strain induced by rMeV-preFS in hamsters were 116.6 and 173.5 (Figure 5F), respectively, representing a 28.3- and 23.5-fold (Figure 5F) reduction compared with those against the BA.2 strain on days 28 and 35 after booster immunization. The GMTs against XBB.1.16 induced by rMeV-preFS in hamsters were 131.8 and 210.7 (Figure 5G), respectively, representing a 22.6- and 19.3-fold (Figure 5G) reduction compared with those against the BA.2 strain after booster immunization. However, the serum conversion rates induced by rMeV-preFS against prototype and variant XBB.1.16 were 100%. It was observed that the serum positive conversion rates in terms of GMTs against the prototype strain and XBB.1.16 for the rMeV-S+SPD-immunized group were 37.5 and 50%, respectively, on days 28 and 35 post-booster immunization (Figure 5F,G). Interestingly, the NAbs titer against the measles virus of serum in the rMeV-preFS- or rMeV-S+SPD-immunized group did not decrease significantly compared with the parental MeV-immunized at three time points (Figure 5H). The results showed that rMeV-preFS displayed higher immunogenicity than rMeV-S+SPD and could also produce cross-reactive antibodies to variants.

### 3.6. rMeV-preFS Vaccine Reduced Virus Loads following Homogeneous Omicron BA.2 Challenge in Golden Syrian Hamsters

To assess the effect of protection in vivo elicited by the MeV-S recombinants, five hamsters in each group were vaccinated at the same dose of 1 × 10^6^ CCID_50_ with parental rMeV, rMeV-preFS, and rMeV-S+SPD via the intranasal immunization route. The animals were challenged at a dose of 1 × 10^6^ TCID_50_ with homogeneous Omicron BA.2. The body weight of each hamster was recorded daily post-challenge. Hamsters in each group challenged with Omicron BA.2 did not exhibit significant weight loss compared with animals in the normal control group (Figure 6A), indicating good tolerance post-immunization and post-challenge.

On day 5 post-challenge, an average titer of 1.58 × 10^2^ CCID_50_/mL and the expression of N protein of Omicron BA.2 was detected in the lungs of animals of the rMeV control group (Figure 6B and Appendix A). Importantly, infectious viruses and the expression of N protein of Omicron BA.2 were not detected in the rMeV-preFS and rMeV-S+SPD group in the lungs, which were significantly lower than that of the rMeV group (Figure 6B and Appendix A). No infectious viruses were detected in the nasal turbinates of the rMeV, rMeV-preFS, and rMeV-S+SPD groups due to the lower sensitivity of virus titration (Figure 6C).

To further evaluate the viral loads, a more sensitive copy number detection of the N and 1ab gene fragments was performed. The Omicron BA.2 genome RNA was examined in lung and nasal turbinate using primers annealing to the ORF1ab and N genes located at the 5′ and 3′ ends of the Omicron BA.2 genome.

Firstly, the levels of total viral RNA, including genomic and subgenomic RNA, were determined using primers annealing to the N gene located at the 3′ end of the genome. The patterns of total RNA titers in the lung (Figure 6D) and nasal turbinate (Figure 6E) were similar to the genomic RNA in the same tissues on day 5 post-challenge. The genomic RNA copies of the N gene in lungs of the rMeV-preFS and rMeV-S+SPD groups were significantly lower than that in the rMeV group (*p* < 0.0001) (Figure 6D). Secondly, the genomic RNA copies of 1ab gene in lungs of the rMeV-preFS group were significantly lower than that in the rMeV-S+SPD group (*p* < 0.01) (Figure 6F). Importantly, the number of RNA copies in lungs from the rMeV-preFS group and rMeV-S+SPD group were both below the detection limit (Figure 6F), whereas the nasal turbinate had RNA titers of 7.4 × 10^4^ and 1.1 × 10^6^, respectively (Figure 6G). Only the number of genomic RNA copies in the nasal turbinate from the rMeV-preFS group were significantly lower than that in the rMeV group (Figure 6G).

In short, these results suggest that rMeV-preFS vaccination intranasally could protect against Omicron BA.2 infection in hamsters by reducing virus loads.

### 3.7. rMeV-preFS Vaccination Protected Hamsters from Omicron BA.2-Induced Lung Pathology

Lungs from all hamsters vaccinated with the two recombinants and challenged with Omicron BA.2 were stained with hematoxylin-eosin, and the severity of histological changes was scored blindly by a staff from the Emerging Infectious Diseases Laboratory. On day 5 post-challenge, lungs from the rMeV group developed extremely severe lung histopathological changes (average score of 6.6) characterized by parenchymatous change, extensive inflammatory cells infiltration, local alveolar edema, and congestion in a large area (Figure 7A,B). Lung pathology in the rMeV-S+SPD group was moderate (average score of 4.0) and significantly less severe than that in the rMeV group (*p* < 0.01) (Figure 7A,B). As expected, the pathological changes in the lung tissues of the rMeV-preFS group were milder than those in the rMeV group (*p* < 0.0001) (Figure 7A,B). Moreover, the score of pathological changes in the rMeV-preFS group was significantly lower than that in the rMeV-S+SPD group (Figure 7B). No lung pathology was detected in the normal control group (score of 0) (Figure 7A,B). These results demonstrated that rMeV-preFS vaccination better protected hamsters from pathological changes in the lungs.

## 4. Discussion

In this study, two highly efficacious MeV-based SARS-CoV-2 Omicron BA.2 variant vaccine candidates were developed, an S chimeric version and an S secretory version, for comparison. It was found that the codon-optimized TM/CT of Omicron BA.2 maintained the function of assisting the S protein to anchor on the surface of measles virus particles, which was confirmed by the sucrose density gradient experiment. Early studies showed that the Newcastle disease virus (NDV) vectors were engineered to express a polybasic cleavage site, removed spike fused to the TM/CT of F, and exhibited superior incorporation into NDV particles, which could potentially be used as an inactivated virus vaccine as well [33]. Moreover, the F protein is in the form of a homotrimer that assists the assembly of the S protein and makes it display on the surface of NDV virus particles in the form of homotrimers, thus improving the immunogenicity of the vaccine [34]. Similar research was performed in rabies virus or VSV (vesicular stomatitis virus) vector-based vaccines. Ebola virus GP protein or SARS-CoV-2 S1 proteins were designed to express fusion with the TMD/CT of G proteins for use as an inactivated vaccine or replicated live virus vector vaccine to induce a protective effect in mice and NHP [35,36,37,38]. In this study, we found that the codon-optimized TM/CT of Omicron BA.2 maintained the function of assisting the S protein to anchor on the surface of measles virus particles, which was confirmed by the sucrose density gradient experiment. Additionally, it was found that TM/CT can help S protein form monomer, dimer, and trimer forms, but the SPD trimer-tag protein can only cause S protein to form monomers and trimers. The possible reason may be that the SPD dimer lacks stability and the low proportion of dimers makes it difficult to be detected.

Measles virus is a non-segmented, negative-sense RNA virus and has been developed as a vector to deliver foreign antigens for preventing some sudden and severe infectious diseases that are extremely difficult to cultivate. So far, more than 100 antigens have been expressed by MeV, and over 20 MeV-based vaccines have been tested in phase I or II clinical trials, including HIV, Zika virus, and CHIKV [39]. However, the rMeV-based SARS-CoV-2 vaccine platform (V591) developed by Merck was announced to be discontinued in phase I and I/II clinical trials because of the insufficient immune responses and seroconversion rate [40]. Thus, they concluded that V591 was generally well tolerated but the immunogenicity was not sufficient for further development [41]. In addition, rMeV-preFS induced high levels of immune response compared with rMeV-S+SPD in hamsters. One possible reason is that preFS protein has TM/CT from Omicron BA.2, which promotes the conformation of trimers closer to natural S protein, while the SPD tag helps in the conformation of S+SPD, which may be different from that of natural S protein. These may lead to the differences in the conformation epitopes of the two kinds of vaccines, as well as the immune responses in hamsters. Therefore, rMeV-preFS is preferred to rMeV-S+SPD as a candidate vaccine.

The advantage of an MeV-based SARS-CoV-2 vaccine is that it can be intranasally administered; however, the subcutaneous route is currently used for MeV vaccination in adults and infants. Early studies suggested that intranasal administration elicits a local immune response, including secretory IgA (sIgA) antibodies providing protection near or at the site of infection of respiratory pathogens [42]. Moreover, respiratory mucosal antibodies are believed to be vital for the protection against the establishment of viral infection before viral exposure or after vaccination [43]. Additionally, the SARS-CoV-2 Omicron variant easily escapes both infection-elicited and vaccine NAbs in the blood [44,45]. It is currently unclear whether efficient mucosal NAbs responses can be induced by vaccination to protect against SARS-CoV-2 infection [46]. Interestingly, intranasal vaccination of rMeV-preFS produced high mucosal sIgA similar to those administered subcutaneously, including the BALF and NAL. However, intranasal inoculation of rMeV-S+SPD induced higher mucosal sIgA compared with subcutaneous vaccination. It may be due to the different forms of antigens, and it shows that the chimeric antigens have more potential as excellent antigens.

The combined vaccine based on attenuated measles, mumps, and rubella viruses (MMR) is one of the most successful vaccines in the history of human epidemic prevention so far. Based on the Centers for Disease Control and Prevention data, one and two doses of MMR vaccine are 93% and 97% effective against MeV. Early research suggested that natural immunity to SARS-CoV-2 might not exist for a long time [47]. However, the MeV vaccine produced long-lasting immunity and protection against MeV infections [48]. In addition, the level of NAbs against MeV did not decrease after inserting the spike genes in the junction between H and L genes compared with the parental MeV-immunized in our study. The MeV-based vaccine has the potential to produce long-term immunity against MeV and other diseases as a combination vaccine. The component of the MeV vaccine was replaced by incorporating rMeV-preFS into the existing MMR vaccine to develop a quadruple vaccine (SARS-CoV-2/MMR) against four important childhood pathogens. Early research suggested that the rMuV-preS-6P vaccine candidate also induced robust humoral and mucosal immunity with or without preexisting MuV antibodies via the intranasal route [49]. Thus, the attenuated MMR has the potential to be further developed into a pentavaccine to prevent five kinds of pathogens in children and has a broad and bright application prospect.

The limitations and shortcomings in our research were that there was no further study on the cellular immune bias of MeV-based candidate vaccines in hamster models. There is a lack of immunological reagents and monoclonal antibodies to study the cellular immunity in the hamster model. Previous studies have reported that commercial anti-mouse mAbs have a certain degree of cross-reactivity for hamsters [50]. However, the cross-reactivity of these monoclonal antibodies to the spleen in hamsters that we tested was too low to support further experiments.

## Figures and Tables

**Figure 1 viruses-16-00559-f001:**
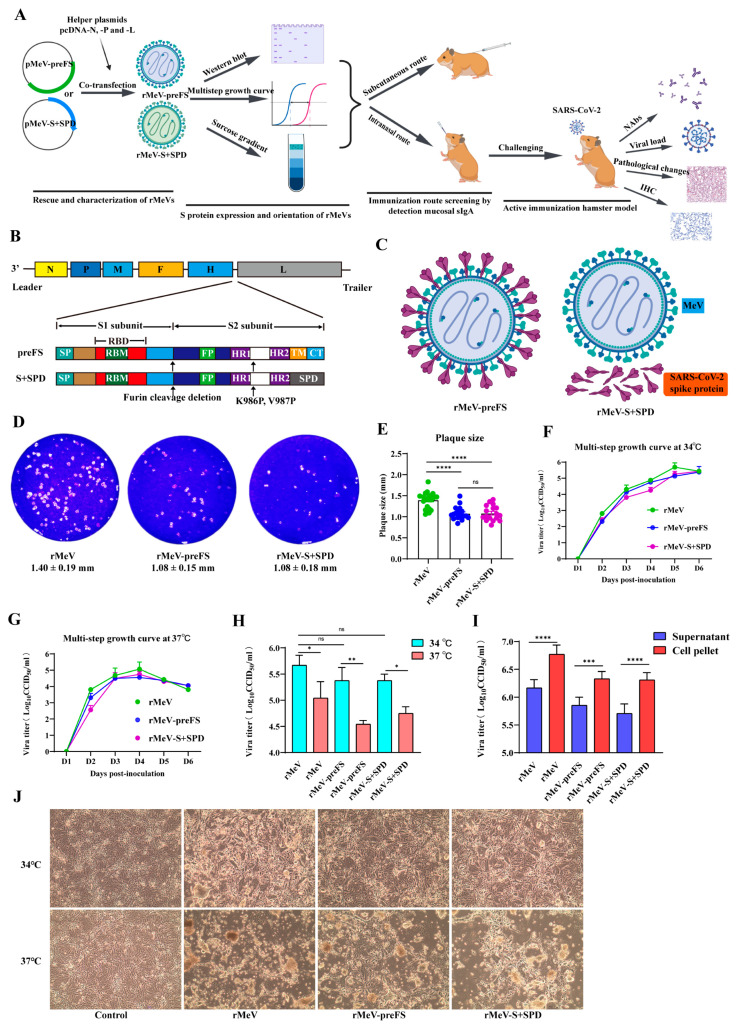
Rescue and characterization of rMeV expressing the stabilized prefusion full-length spike or secretory ectodomain of SARS-CoV-2. (**A**) Experimental flow chart. (**B**) Strategy for insertion of preFS and S+SPD of Omicron BA.2 into the MeV genome. The stabilized, prefusion preFS and S+SPD genes were inserted into the gene junction between the H and L genes in the genome of the MeV-S191 vaccine strain. The domains of the spike protein are shown. SP, signal peptide; RBM, receptor binding motif; RBD, receptor binding domain; HR, heptad repeat; FP, fusion peptide; CH, central helix; CT, cytoplasmic tail; TM, transmembrane domain; SPD, human pulmonary surfactant protein-D sequence. The organization of the genes in the negative-sense MeV genome is shown. N, nucleoprotein gene; P, phosphoprotein gene; M, membrane protein gene; F, fusion protein gene; H, hemagglutinin gene; L, large polymerase gene. (**C**) The expression diagram of SARS-CoV-2 spike proteins in rMeV-preFS- and rMeV-S+SPD-infected cells. (**D**) The plaque morphology of rMeV expressing preFS or S+SPD proteins and parental strain were observed on day 6 post-infection. (**E**) The average diameters of 18 randomly selected plaques from each measles virus were compared. (**F**,**G**) Multistep growth curves were determined in confluent monolayers of Vero cells in six-well plates. Cells were infected with individual viruses at a multiplicity of infection (MOI) of 0.001 at 34 °C (**F**) or 37 °C (**G**). From day 1 to day 6, viral titers in supernatants were determined by a CCID_50_ assay. (**H**) The peak titers in supernatants of individual recombinants within 6 days at 34 °C or 37 °C were compared. (**I**) The titers in supernatants and cell pellets of individual recombinants were titrated in Vero cells in T25 flasks infected at an MOI of 0.001 at 34 °C. The infected cells were frozen and thawed twice, and cell pellets were resuspended in DMEM in the same volume as the harvested supernatants. (**J**) The morphology of syncytia formed by rMeVs at 34 °C or 37 °C was observed on day 5 post-infection. Statistical assay: *, *p* < 0.05; **, *p* < 0.01; ***, 0.001; ****, *p* < 0.0001; ns, not significant.

**Figure 2 viruses-16-00559-f002:**
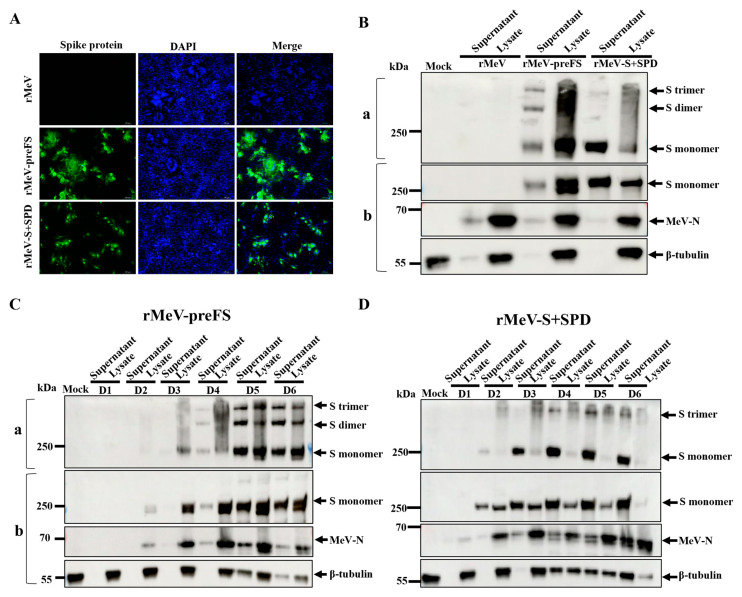
Analysis of spike proteins expressed by rMeVs using IFA and Western blot assays. (**A**) Analysis of preFS and S+SPD protein expression by IFA. Vero cells in 6-well plates were infected with each virus at an MOI of 0.001 at 34 °C. At 72 h post-infection, the cells were fixed with 80% pre-cooled acetone and stained with anti-SARS-CoV-2 RBD chimeric mAb. (**B**) Analysis of Omicron BA.2 preFS and S+SPD protein expression in supernatants and cell lysates by Western blotting. Vero cells in T25 flasks were infected or mock-infected with individual recombinants at an MOI of 0.001 at 34 °C. At 96 h post-infection, medium supernatants were removed and cells were lysed in 500 μL of SDS lysis buffer and diluted tenfold to the same volume of the harvested supernatants. Proteins in cell lysates and supernatants were treated without BME (**a**) or with BME (**b**) and separated by SDS-PAGE. (**C**,**D**) Time courses of preFS protein (**C**) and S+SPD protein (**D**) expression in cells and media by Western blotting. The same volumes of lysates or supernatants harvested on days 1, 2, 3, 4, 5, and 6 post-infection were analyzed. Membrane-blotted proteins were probed with mouse anti-Omicron RBD protein monoclonal antibody, rabbit anti-MeV-N3 antiserum, and mouse anti-β-tubulin antibody in a Western blotting assay.

**Figure 3 viruses-16-00559-f003:**
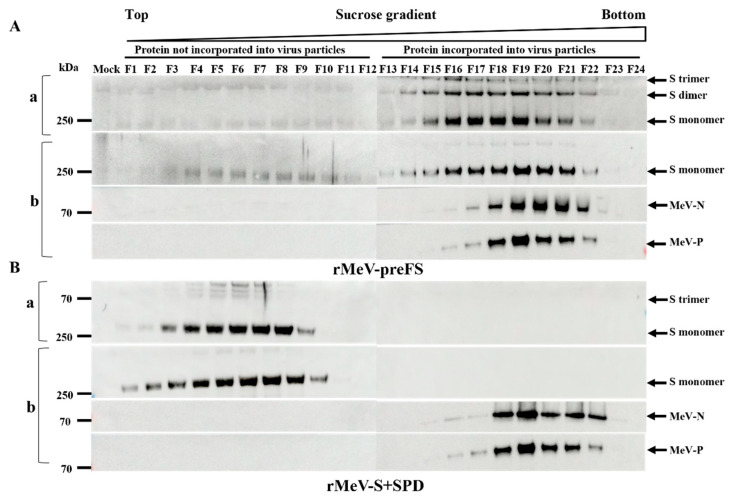
Expression analysis of spike proteins by rMeVs through 10–60% sucrose density gradient. Vero cells were infected with the recombinants at an MOI of 0.01 at 34 °C, and supernatants were harvested on day 4 post-incubation. The supernatants were centrifuged through a 10–60% sucrose gradient. The viral proteins in individual fractions derived from supernatants of infected cells with rMeV-preF S (**A**) and rMeV-S+SPD (**B**) were treated without BME (**a**) or with BME (**b**) and separated by SDS-PAGE and analyzed by Western blotting. Anti-Omicron RBD monoclonal antibodies, rabbit anti-MeV-N3 antiserum, and anti-MeV-P antibodies were used for the detection of viral proteins, respectively.

**Figure 4 viruses-16-00559-f004:**
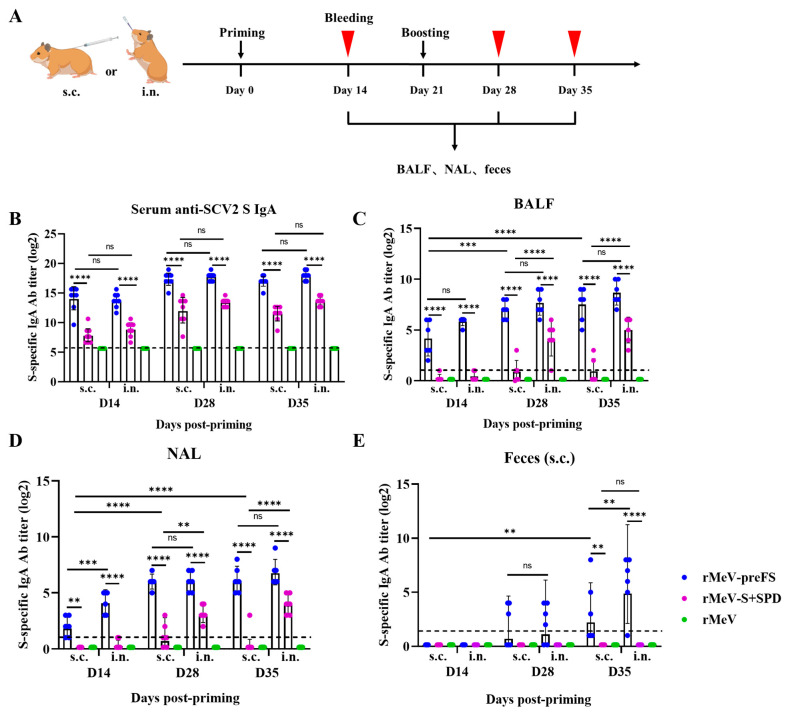
Mucosal immune responses in immunized golden Syrian hamsters by rMeV-preFS and rMeV-S+SPD through intranasal and subcutaneous routes. (**A**) Immunization schedule of golden Syrian hamsters. Groups of four-week-old female hamsters (n = 6) were immunized intranasally (i.n.) or subcutaneously (s.c.) at a dose of 1 × 10^6^ CCID_50_ of parental rMeV, S recombinants rMeV-preFS, and rMeV-S+SPD or mock-immunized with DMEM. Hamsters were boosted 3 weeks post-priming. (**B**) IgA levels in sera via s.c. and i.n. routes. (**C**) IgA levels in the BALF via s.c. and i.n. routes. (**D**) IgA levels in the NAL via s.c. and i.n. routes. (**E**) Fecal IgA levels via s.c. and i.n. routes. Samples were collected on days 14, 28, and 35 post-priming. Data were analyzed using a two-way ANOVA test (**, *p* < 0.01, ***, *p* < 0.001; ****, *p* < 0.0001, ns, not significant) and presented as the GMT of six hamsters ± SD.

**Figure 5 viruses-16-00559-f005:**
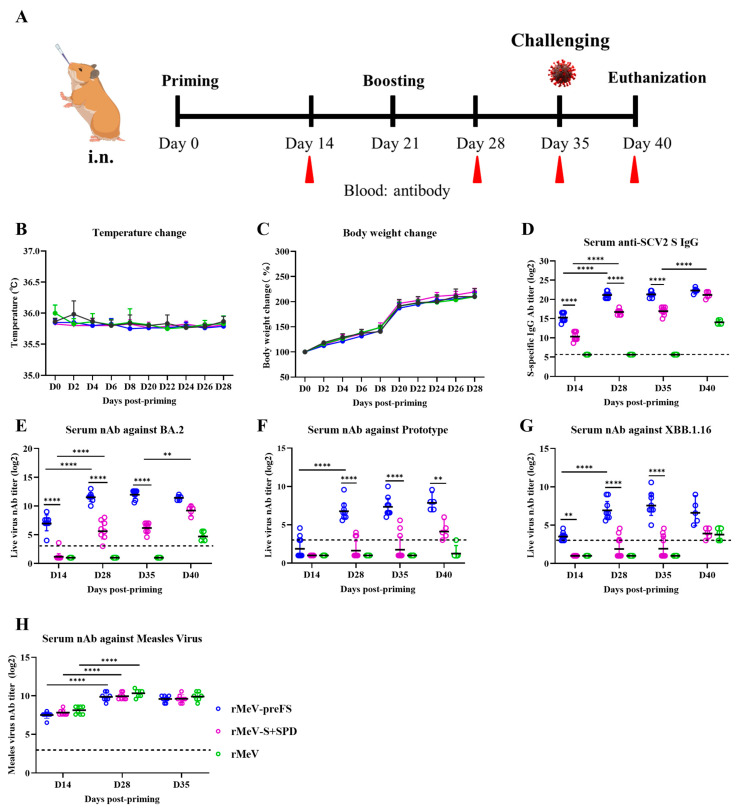
Tolerance and immunogenicity of rMeVs expressing Omicron BA.2 spikes in golden Syrian hamsters against different strains. (**A**) Immunization schedule of golden Syrian hamsters. Four-week-old female hamsters (n = 8) were immunized intranasally with 1 × 10^6^ CCID_50_ of parental rMeV, rMeV-preFS, or rMeV-S+SPD or mock-immunized with DMEM and boosted 3 weeks post-priming. On days 14, 28, and 35, the sera were collected for antibody detection. On day 35, hamsters were intranasally challenged with 1 × 10^6^ CCID_50_ of Omicron BA.2. Unimmunized, unchallenged animals in the control group were inoculated with DMEM. (**B**,**C**) Evaluation of tolerance of each recombinant post-priming. The temperatures (**B**) and changes in body weight (**C**) of individual hamsters were monitored at intervals of two days post-priming. (**D**) Serum ELISA-specific IgG levels against Omicron BA.2 S protein. (**E**) Serum NAbs levels against Omicron BA.2 strain. (**F**) Serum NAbs levels against SARS-CoV-2 prototype strain. (**G**) Serum NAbs levels against SARS-CoV-2 XBB.1.16 strain. (**H**) Serum NAbs levels against measles virus. Data are the GMT of eight animals ± SD pre-challenge or five animals ± SD post-challenge. Data were analyzed using a two-way ANOVA test. (**, *p* < 0.01; ****, *p* < 0.0001).

**Figure 6 viruses-16-00559-f006:**
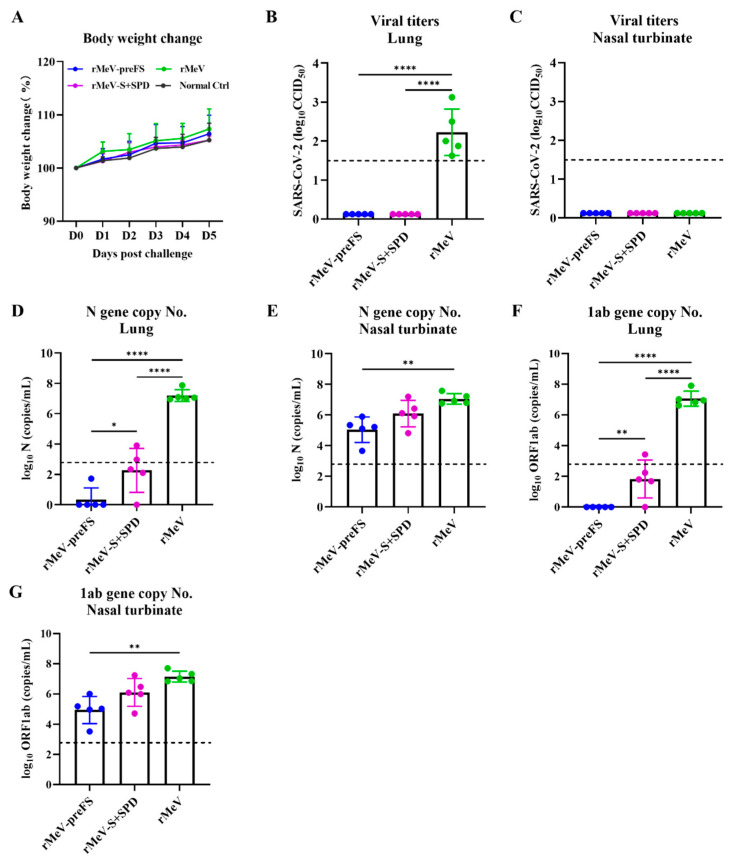
Body weight changes and viral loads post-challenge against Omicron BA.2 in golden Syrian hamsters vaccinated by rMeV-preFS and rMeV-S+SPD via the intranasal immunization route. (**A**) Dynamics of hamster body weight changes post-challenge with Omicron BA.2. The body weight for each hamster was recorded from day 0 to day 5. On day 5 post-challenge, all hamsters were euthanized. The viral titers of Omicron BA.2 in the lungs (**B**) and nasal turbinate (**C**). Viral titers are the GMT of five animals ± SD. The limit of detection (LoD) is 2.7 log_10_ CCID_50_/mL per gram of tissue (dotted lines). (**D**,**E**) Omicron BA.2 subgenomic RNA copies of the N gene in lungs (**D**) and nasal turbinates (**E**). (**F**,**G**) Omicron BA.2 genomic RNA copies of the ORF1ab gene in lungs (**F**) and nasal turbinates (**G**). The dotted lines indicate the limit of detection. Data were analyzed using a two-way ANOVA test (*, *p* < 0.05; **, *p* < 0.01; ****, *p* < 0.0001).

**Figure 7 viruses-16-00559-f007:**
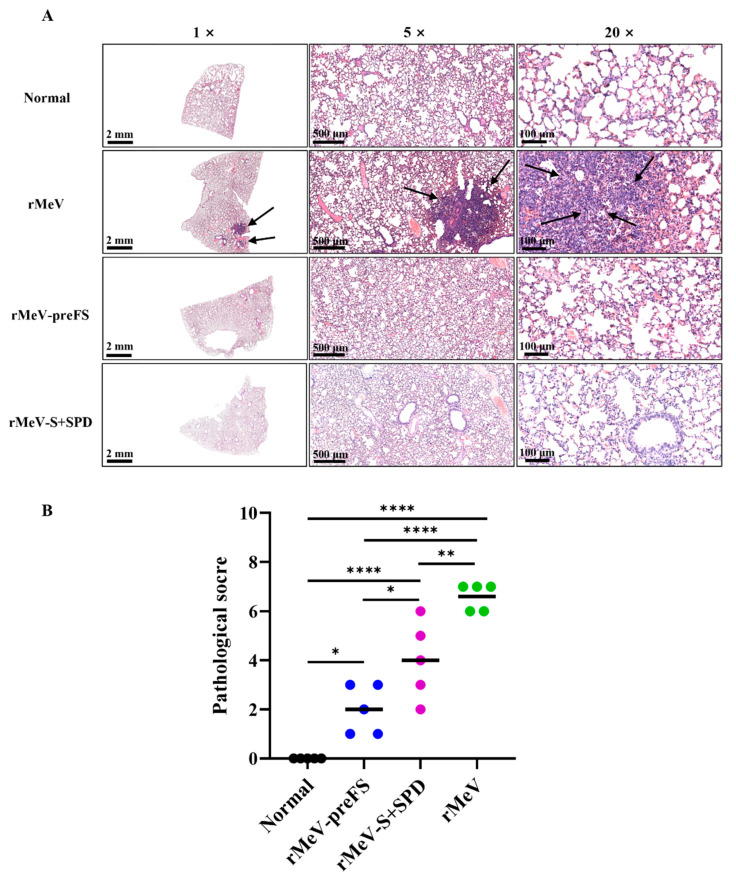
Lung pathology and virus replication post-challenge of Omicron BA.2 for hamsters immunized with rMeV-preFS and rMeV-S+SPD. (**A**) The lung tissues were fixed and embedded in paraffin on day 5 post-challenge. Following samples being sectioned, deparaffinized, and rehydrated, hematoxylin-eosin staining was performed for the examination of histological changes under microscopy. Lung histopathological changes were indicated by arrows. Micrographs of 1×, 5×, and 10× magnifications of a representative lung section from each group were shown, and scale bars were indicated at the left corner of each image. (**B**) Lung pathology scores post-challenge with Omicron BA.2. Each slide was evaluated based on the severity of histological changes, including edema, hyaline membrane, fragments of necrotic cell, neutrophil infiltration, monocyte, and thrombus. Scores 6–8, extremely severe pathological changes; scores 4–6, severe pathological changes; scores 2–4, moderate pathological changes; scores 0–2, mild pathological changes; score 0, no pathological changes. Data were analyzed using a two-way ANOVA test (*, *p* < 0.05; **, *p* < 0.01; ****, *p* < 0.0001).

## Data Availability

Data are contained within the article and Appendix A.

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
