# Peer review of "Measles Virus-Based Vaccine Expressing Membrane-Anchored Spike of SARS-CoV-2 Inducing Efficacious Systemic and Mucosal Humoral Immunity in Hamsters"

_viruses, 2024, doi:10.3390/v16040559_

Round 1

Reviewer 1 Report

Comments and Suggestions for Authors

The authors have presented a robust series of experiments detailing the development and testing of SARS-CoV-2 vaccines utilizing the measles vector, incorporating the S protein in its prefusion trimeric form, either as membrane-anchored or preferentially secreted trimer. They provide thorough in vitro characterization alongside virological assessments and meticulously conducted controlled in vivo experiments. However, my primary critique lies in the paper's length, lack of focus, and absence of clear conclusions. Instead of offering concise conclusions, the paper tends towards comparative analyses. For instance, the authors fail to hypothesize which vaccine candidate they favor (preFS, S+SPD) and under what circumstances, as well as neglect to discuss the factors driving observed in vivo differences. Additionally, although they comment on the decline of neutralization antibodies/narrow focus with other vaccines, they omit discussing whether this trend applies to their vaccine. Thus, the authors should explicitly articulate their conclusions and assumptions rather than merely presenting observations.

 My secondary concern pertains to the abstract, where some statements do not entirely align with their results. For example, the assertion that 'The intranasal delivery of vaccine candidates induced a more robust mucosal IgA antibody response than the subcutaneous route' is accurate for only one vaccine candidate (S-SPD). Furthermore, they fail to directly compare the routes on the same graph, although this comparison is implicit for the S+SPD candidate. Additionally, they must decide which data to prioritize; for instance, the pathology presented in Figure 7 results from viral protein expression in tissues (Figure 8), suggesting that one of these images could be relegated to the supplement. The entire second part of the manuscript must be shortened, including the discussion.

 Moreover, the authors should carefully use terms like 'native,' 'denatured,' 'reduced,' and 'nonreduced,' particularly in the context of lanes 333 to 337 in Figure 2B, which require revision. Figure 2B lacks conviction compared to 2C, where the absence of the dimer form is more evident. Additionally, this dimer issue is not explicitly explained. Minor issues include typos such as 'rMV3' (line 286), 'strained' (line 361), missing labels for fluorescence in Figure 2A, labeling 4a instead of 5a in line 415, the 'Meales' virus typo in panel 5H, etc. As a suggestion, Figure 4 could benefit from using 2 panels in a row (instead of 3) to facilitate direct parameter comparison.

In conclusion, while the paper presents commendable and substantial data, its presentation requires refinement to effectively convey the message to the audience. This entails shortening and delivering a more concise narrative as well as choosing essential data to keep the readers engaged.

Reviewer 2 Report

Comments and Suggestions for Authors

Reviewer Commente to the authors of the manuscript intitle "Measles virus-based vaccine expressing membrane-anchored spike of SARS-CoV-2 inducing efficacious systemic and mucosal humoral immunity in hamsters" by Zhi-Hui Yang et al.

The manuscript is interesting and proposes a new model using principles already explored in other vaccines. By improving the exposition it will certainly be interesting for readers to appreciate the scientific contribution despite the important limitations that the authors themselves recognize.

Minor comments

- In the introduction section there is an error, after the aim of the study start at the line 85, perhaps it is a refuse: "MeV vaccine candidates have been found to induce a wide range of mucosal immune responses intranasally, better than those administered subcutaneously, and provide a complete protection against vaccines Omicron BA.2 in animal models.". Please clirify.

- To help readers better understand the experimental setup and phases, it would be appropriate to illustrate the sequence of experimental phases using a flowchart.

- In the results section, the first point (from line 271 to line 277) describes the experiment, it is better to insert it in the Materials and Methods section.

-
